# A How to Guide: Clinical Population Test Development and Authorization of MALDI-ToF Mass Spectrometry-Based Screening Tests for Viral Infections

**DOI:** 10.3390/v14091958

**Published:** 2022-09-03

**Authors:** Ray K. Iles, Jason K. Iles, Raminta Zmuidinaite, Michael Roberts

**Affiliations:** 1MAP Sciences Ltd., The iLAB, Stannard Way, Priory Business Park, Bedford MK44 3RZ, UK; 2Laboratory of Viral Zoonotics, Department of Veterinary Medicine, Cambridge University, Madingley Road, Cambridge CB3 0ES, UK; 3Chem Quant Analytical Solutions, LLC, 1093 Investment Blvd, Apex, NC 27502, USA

**Keywords:** COVID-19, SARS-CoV-2, viral infection, detection, asymptomatic screening, MALDI-ToF MS, CLIA-LDT, clinical process

## Abstract

Applying MALDI-ToF mass spectrometry as a clinical diagnostic test for viruses is different from that of bacteria, fungi and other micro-organisms. This is because the systems biology of viral infections, the size and chemical nature of specific viral proteins and the mass spectrometry biophysics of how they are quantitated are fundamentally different. The analytical challenges to overcome when developing a clinical MALDI-ToF mass spectrometry tests for a virus, particularly human pathogenic enveloped viruses, are sample enrichment, virus envelope disruption, optimal matrix formulation, optimal MALDI ToF MS performance and optimal spectral data processing/bioinformatics. Primarily, the instrument operating settings have to be optimized to match the nature of the viral specific proteins, which are not compatible with setting established when testing for bacterial and many other micro-organisms. The capacity to be a viral infection clinical diagnostic instrument often stretches current mass spectrometers to their operational design limits. Finally, all the associated procedures, from sample collection to data analytics, for the technique have to meet the legal and operational requirement for often high-throughput clinical testing. Given the newness of the technology, clinical MALDI ToF mass spectrometry does not fit in with standard criteria applied by regulatory authorities whereby numeric outputs are compared directly to similar technology tests that have already been authorized for use. Thus, CLIA laboratory developed test (LDT) criteria have to be applied. This article details our experience of developing a SAR-CoV-2 MALDI-ToF MS test suitable for asymptomatic carrier infection population screening.

## 1. Introduction

For the early detection of asymptomatic and pre-symptomatic viral infection, reliance is on PCR technology. Immunoassay based lateral flow test kits (LFTs) are widely available, but their ability to catch the disease early, and therefore effect a public health containment of outbreaks effectively, is a highly contentious debate [1,2,3].

The authorized rtPCR and similar LAMP based tests are specific for SARS-CoV-2 strains and are highly sensitive. However, the same test(s) cannot be used for the detection of a new virus or could give false negatives when variants arise., i.e., if a base pair mutation destroys a primer recognition sequence, as has happened for SARS-CoV-2 Spike protein gene sequence primers for the Delta and Omicron variants. In addition, rtPCR and LAMP technology is not easily adaptable to measure viral load as an affordable option.

The economics of testing a population on such huge scales are staggering; for PCR-based tests, the cost are high (up to GBP 150 for individuals), and, worryingly, they are dependent on other countries for supplies of multiple component reagents.

Lateral flow devices/tests (LFTs) are considerably cheaper (although over GBP 4 billion of the UK Tax budget has been spent on them) and easier to use, even allowing home testing by the general public. However, sensitivity is an issue. This has given rise to questions about inadequate sampling, and when combined with the reduced sensitivity compared to PCR, whether its use should be confined to being only a definitive or confirmatory test of infection in those with clinical symptoms of an upper respiratory tract infection (i.e., pyrexia, fatigue, cough, loss of sense of smell and cold/flu-like symptoms). LFTs’ limited ability to reliably detect infection in asymptomatic carriers or most pre-symptomatic infections is their greatest weakness. Thus, their widespread use as a population-based test to prevent contact spread is questionable [4,5,6,7].

Therefore, the clinical need is for a home sampled test where the efficiency of sampling can be confirmed, that is as sensitive as PCR but not so specific that variant mutations, or indeed other emerging viral infections, could not be simultaneously detected; that is cheaper and less reagent- and skilled operator-dependent; and that is fast and identifies asymptomatic carriers.

In our development of a MALDI-ToF mass spectrometry test for enveloped viruses, in particular for SARS-CoV-2, we have attempted to address all these challenges.

### 1.1. Understanding the Systems Biology of the Virus and Viral Infections

From a clinical analytical point of view, the first consideration is what molecules to look for. Ideally, these are going to be specific and unique to the pathogens. With MALDI-ToF analysis of bacteria, this is the detection of abundant and species-specific ribosomal and other associated housekeeping proteins. These are relatively small and simple poly-peptides (e.g., not glycosylated) that can be resolved easily and with high definition by MALDI-ToF mass spectrometry. Furthermore, bacteria can be grown on artificial media and sampled as near pure colonies without contamination from other proteins arising from the host biological sample. The first hurdle in a viral mass spectrometry test is that no equivalent housekeeping metabolic proteins exist for viruses. Indeed, the main feature is that they cannot be propagated upon artificial media in vitro and therefore enriched in quite the same way. Propagation has to be on a host tissue cell culture. Although a number of unique proteins are expressed by hijacking the host cells housekeeping and protein synthesis systems, the only stage at which a host cell protein contamination-free sample can be found is when the virion particle is released, allowing isolation from the host cells, or fragments of host cells.

Virion particles contain a limited number of distinctive proteins. Functionally, they do not require any of the multitude of proteins that sustain the characteristics of living organisms other than the ability to replicate. In this respect, the only proteins required are those that protect its genome (nucleocapsid) and those that enable it to infect other cells such as spike/receptor glycoproteins (see Figure 1).

Thus, in mass spectrometry-based detection assays, viral proteins will only be a minor component and masked amongst a forest of infected host tissue/cell proteins, but in a virion-targeted or -enriched fraction, the viral-specific proteins, although more limited, have a much higher chance of being resolved and visualized (Figure 2). Other techniques can overcome the problem of being lost within a crowd of other proteins. For example, the resolution power of liquid chromatographic (LC)-coupled ESI–mass spectrometry is extremely good, and resolving characteristic molecules, even within a crowded forest situation, is much better by LC–MS than by a MALDI-based system. However, this is limited to proteins of less than 2–5 kD, and so time-consuming and technically more demanding tryptic digestion of a sample is required before mass spectral analysis [8,9,10].

### 1.2. Understanding the Nature of Viral Proteins and Molecular Biology

A distinguishing feature of the unique proteins of a virus, whether expressed as part of a virion particle or the infective process within the host cell, is their large size. The advantage of MALDI-ToF mass spectrometry is the huge dynamic range of masses it can resolve and its ability to ionize large proteins intact with only one or two charges (z). This is something LC–ESI-based mass spectrometry cannot achieve, particularly in a complex biological mixture [10].

Having said this, even with MALDI–ToF mass spectrometry, you have to consider the nature and characteristic of the target proteins. The first consideration is given to the matrix needed to give charge. In microbiology, alpha-Cyano-4-hydroxycinnamic acid (CHCA) is the standard and almost universal matrix for analysis of small molecules and largely unmodified peptides/small proteins of up to approximately 10,000 Daltons in mass [11,12].

However, CHCA is not the most appropriate matrix for virion particle analysis: nucleocapsid proteins are much larger proteins (>10 kD) and form a complex tessellation polymeric shells to protectively encase, whilst also holding in a stable formation, the nucleic acid viral genome. For non-enveloped viruses, within this capsid tessellation are also specific protein elements that effect infection of the target host cells (see Figure 1). The protein components of the nucleocapsid tend to carry phosphorylation and other post-translational modification, such as covalent linkages with small ubiquitin-like modifier proteins (SUMOylation) [13,14]. This increases their final mass as determined by a mass spectrometer and adds variability/peak spreading around that mass peak maxima.

**Figure 2 viruses-14-01958-f002:**
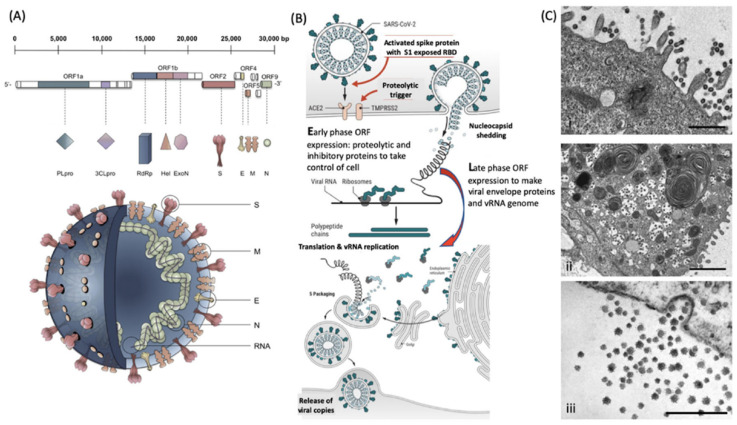
Molecular cellular pathology of SARS-CoV-2 infection. Panel (**A**). Schematic illustration of SARS-CoV-2 viral RNA genome in relation to order of expression and functional incorporation with the virion particle. The genome comprises a 5′-untranslated region (5′-UTR), open-reading frames (ORFs) 1a and 1b encoding nonstructural proteins 3-chymotrypsin-like protease (3CLpro), papain-like protease (PLpro), helicase (Hel), and RNA-dependent RNA polymerase (RdRp) as well as accessory proteins, and the structural S protein (S), E protein (E), M protein (M), and Nucleocapsid protein (N) in ORFs 2, 4, 5 and 9. Panel (**B**). Schematic of virion fusion mediated by Spike protein attachment to the ACE2 receptor and accessory cleavage of the Spike protein quaternary complex, resulting in fusion peptide exposure and binding to target cell plasma membrane. Focal, host cell and viral envelope lipid bilayer membrane fusion and viral RNA ingress of the target host cell. Shedding of nucleocapsid and early ORF expression to take control of the cell and inactivate internal anti-viral proteins (e.g., Ubiquitin and RNase). Late ORF expression: replication of viral RNA genome and expression of, via post-translational processing mechanisms of the endoplasmic reticulum and Golgi apparatus (e.g., glycosylation), membrane embedded proteins that are responsible for infectivity including the Spike protein complex. Copies of SARS-CoV-2 vRNA are packaged into lipid bilayer membrane envelopes containing multiple copies of S, E, M and N, glyco- and phospho-proteins. Inclusion transport vesicle, packed with enveloped virions, fuses with host cell membrane releasing infective SARS-CoV-2 virion. Panel (**C**). Electro-micrographs of multiple virions attacking the cell membrane (**i**), viral genome expression subsuming all functions of the cell to form thousands of virion copies within inclusion vesicles (**ii**) and finally release of multiple virions by fusion of inclusion packaging vesicles with the cell surface plasma membrane of the infected host cell (**iii**) [15]. Original composite figure adapted in parts from Depfenhart et al. [16] Kupferschmidt and Cohen [17] and Mason [18]. Scale bar: 1000 nm.

The variability in mass of characteristic proteins found in the enveloped viruses resolved by mass spectrometry can be very significant. Lipid membrane-embedded proteins of the enveloped viruses undergo post-translation modification by glycosylation and lipid moiety attachments (e.g., palmitoylation). Hydrophobic domains and lipid attachments embed and orientate the external protruding glycosylated regions of these unique viral genome envelope proteins, which mediate the attachment and invasion of the target host cells.

The large and polymeric nature of nucleocapsid proteins can be overcome relatively simply by denaturing and reducing condition that liberate the monomers. Although still large, these require matrices capable of ionization of large predominately water-soluble proteins. Many of these are within the operational mass range of MALDI–ToF mass spectrometers currently certified and used for clinical diagnostics (i.e., bacterial identification). Theoretically, proteins of up to 1 million Daltons can be detected, but in our experience, usually, these instruments are not sufficiently sensitive above 250,000 Daltons (i.e., can detect picomoles of chosen marker protein > 250,000 Daltons on a target plate well).

Additionally, in our experience, Sinapinic acid (SA) is the most versatile matrices for ionization of large post-translationally modified proteins, but this does have its own unique problems compared to CHCA (see operational parameters of chosen mass spectrometer).

The enveloped virus—the equivalent to pirate exosomes—present an added complication of liberation and solubilization from the lipid membrane. The combined combination of size, solubilization and ionization are significant hurdles in sample processing prior to MALDI–ToF mass spectrometry.

## 2. Sampling and Virion Enrichment

The prevailing sampling method for upper respiratory viruses is nasal–pharyngal swabs. The direct application of swabs onto MALDI–ToF mass spectral plates, followed by the addition of a matrix, has been claimed to result in distinctive mass patterns correlating with SARS-CoV-2 infection at masses less than 20,000 m/z [19,20]. As detailed above, it is difficult to imagine that these are arising specifically, and therefore characteristically, from the SARS-CoV-2 virus itself. Indeed, particularly if not chemically modified, the swab sample will contain many resident bacterial and human proteins within this mass range, including inflammatory cytokines. Thus, it is very doubtful that any such associating marker mass spectrometry peaks would be specific for SARS-CoV-2, from the direct application of a swab sample onto a MALDI-ToF plate and addition of CHCA alone, as has been claimed in several studies [19,20]. In these studies, the comparison has been with PCR-positive versus PCR-negative samples only, with complex bioinformatics establishing a peak profile correlating with infection. However, these peaks probably arise due to any upper-respiratory infection and correspond to immune response molecules/small proteins such as cytokines. Thus, specificity is a major issue in this approach.

In order to achieve specificity, some form of selection or enrichment of the virus is required prior to mass spectrometry. In addition, the mass spectrometry technique adopted should be directed against the unique proteins of the virion causing the infectious disease. Clearly, having an in vitro culture step within the clinical test protocol is neither cost-effective nor feasible for mass-testing purposes. There are various approaches to the direct enrichment of virions from a sample. These include highly selective solid phase processes (e.g., magnetic bead conjugated antibodies and aptamers) and the semi-selective precipitation of large particles [21,22]. Being overly selective has disadvantages in that variants or other viruses are not captured. In semi-selective enrichment, all viruses present can be captured and not only detected but identified from the resolved mass spectral pattern. Thus, not only does the test detect a target virus infection but can differentiate, in one sample analysis, the precise virus cause of an infection from the mass spectral pattern obtained. This not only allows for differential diagnosis but mutation causing antigenic drift and zoonosis—new pathogenic virus detection.

In our work on SARS-CoV-2, we found that the simple acetone precipitation of the virions from the sample not only enriched them but also rendered them inert [15].

### 2.1. Virion Protein Solubilisation and Extraction

Protein denaturation in water, or in combination with volatile organic chemicals that evaporate leaving no residues (such as acetonitrile), can effectively denature proteins, whilst high-concentration salt-based denaturation will suppress ionization and/or cause significant adducts with the target proteins. Most reducing agents, such as Dithiothreitol, b-mercaptoethanol and tris(2-carboxyethyl)phosphine, are water-soluble, so complete denaturation and fragmentation of a complex protein structure (whether subunits or proteolytic activated complexes, which are held together by di-sulphide bridges) can be achieved without compromising ionization and hence sensitivity.

The water-based solubilization of hydrophobic proteins and extraction from lipid membranes is usually achieved by the high-concentration addition of detergents and surfactants such as SDS, Triton and CHAPS. However, these significantly inhibit ionization. This has been overcome by the use of a series of non-ionization suppressing surfactants that have been developed by MAPSciences to be used at low concentrations. In our work on enveloped viruses, one termed LBSD-X has proved to be extremely versatile and efficient (see Figure 3).

### 2.2. Accelerating MALDI-ToF Assay Protocol Development Using Pseudotypes/Pseudoviruses

The ability to use and distribute a non-replicating form of virion particle, constructed to express target viral envelope proteins, post-translationally modified in a similar manner to those of the wild-type virus in vivo, allows correct extraction and mass peak determination (see Figure 4). Thus, the use of “pseudo virus” constructs is invaluable in both the initial development and subsequent quality control of MALDI–ToF virus tests [23].

However, live virus testing is an essential final stage of test development, and working with culture fluid containing virion particles will require category 3 facilities and/or robust processing to render the virus inactive. Fortunately, UV irradiation at 280 nm and acetone precipitation are both highly effective in deactivating enveloped viruses, including SARS-CoV-2 [25,26].

In addition, analysis of the biological sample is needed, as residual or co-enriched proteins from the biological sample, such as antibodies, can complicate the spectra and limit what spectral features can be resolved without interference (see Figure 5 and Figure 6). Nevertheless, an increased antibody proteins response can contribute as a biological marker feature of an infection (see Figure 6) [15].

### 2.3. Understanding the Operational Parameters of Your MALDI-ToF Mass Spectrometer

Within the clinical field of analysis, unlike for research instruments, MALDI-ToF mass spectrometers are used continuously. The two critical components that account for deterioration in analytical performance are the accumulation of desorption matter/ionization spray fouling of the source/primary lens, and the deterioration of the detector.

The rate of primary lens fouling is a function of matrix, the sample (i.e., urine, serum/plasma or extract) and how much energy you need to apply from the laser to achieve the ionization of your target molecules within the sample. CHCA, which requires low-energy bursts from the laser, is the least fouling matrix formulation, whilst high-energy bursts to the Sinapinic acid matrix are amongst the most fouling. Thus, the analysis of glycoproteins embedded in lipid-enveloped virion particles has much higher primary lens fouling than the analysis of bacterial house-keeping proteins [15].

An additional problem is detector response to slower moving large biomolecules, many of which may have lost their gained positive charge post initial acceleration by the time they hit the detector at the end of a flight tube. In order to achieve good responses, often, the voltage gain across the detector has to be raised. This shortens the operational life of the detector [27].

Thus, in the operationalization of a MALDI–ToF system for viral detection, the degree to which the primary/source lens can be regularly cleaned and accessed for manual maintenance is an important feature. Furthermore, detector gain will need to be regularly monitored for loss of signal intensity. The primary acceleration voltage will play a part in the longevity of the detector, as current instruments’ initial voltage acceleration values are either set at 15 KV or 20 KV. This dramatically increases voltage gain, so that detectors on instruments need to be set to record large and partially hydrophobic glycoproteins in MALDI–ToF mass spectrometers [15,27,28].

### 2.4. Peak Identification–Quantification and Bioinformatics

Automated and machine learning are key features of mass spectrometry diagnostics of infectious diseases and other human disorders. Peak mass and peak height/intensity are key components fed into bioinformatic routines, direct from a mass spectrometer’s output data. However, unlike the mass spectral peaks that arise from distinct biomolecules and even peptides, such as those characterized in current MALDI–ToF mass spectrometry of bacterial samples, the shape and resolution of viral specific protein peaks is not as precise (see Figure 5). The broadening of the peak is partly because of the sheer number of isotopic combinations of the many hundreds of carbon, hydrogen, and nitrogen ions, which is joined by the even more frequent isotopic variants of sulphur. Of far more significance to our studies of viruses is the variability in mass due to post-translational modifications [29]. For example, N-Linked and O-linked sugar moieties can be mono-, di- and tri-antennary and may or may not be partially degraded and missing terminal residues, depending on enzymes and the chemical environment; sialic acid glycan terminal residues alone are easily lost by acidic extractions [30,31]. At approximately 200 m/z each, the loss of terminal sialic acids from a tri-antennary N-Linked glycan can vary the mass of a single pepto-glycan chain by 600 Daltons.

This alters the mass spectral characteristics from a relatively sharp peak of defined precise mass and peak intensity to a broad asymmetric peak, where a maximum is within a range and peak area. As a result, rather than peak height, peak area has to be calculated to characterize relative intensities (see Figure 7). Thus, the calculation of mass peak characteristic of viral proteins has to allow for post-translational modification and isotopic variation.

Viral nucleocapsid proteins tend not to be glycosylated but to be present with broadened mass spectra by the sheer number of isotopic combinations, ADP-Ribosylation and SUMOlyation (not seen in recombinant-bacterial expression). Enveloped viruses also contain variably post-translationally modified and heavily glycosylated proteins embedded with a lipid membrane [14,32,33,34].

Many of the proteomic bioinformatic tools available have been written to recognise and process sharp intense spectral mass peaks. Thus, modified and new bioinformatics tools are required for large proteins, and particularly large post-translationally modified proteins require mass spectra. These tools need to report peak maxima within a comparatively wide defined range and then calculate the peak area, before applying machine learning for pattern recognition [35,36]. In addition, when applying machine learning to a clinical diagnostic as opposed to a laboratory-based problem in mass spectrometry pattern recognition, a high degree of uncertainty has to be allowed for in the training sets [37].

### 2.5. Promise and Pitfalls of Machine Learning Bioinformatics

The two-dimensional simultaneous measurement of multiple markers, such as that afforded by MALDI–ToF mass spectrometry, needs bioinformatics not only to interpret but to sort the significant spectral features [15]. The ability to match spectral patterns of clinical patient samples to a disease or medical condition with a high degree of accuracy is the goal of clinical mass spectrometry; machine learning systems (MLS) are the way forward. However, although overfitting is a recognized problem, mathematicians need to take into account that nothing is 100% certain, and that is particularly true in the diagnosis of a viral infection. If your machine learning algorithm matches (96–100%) the result of a given pre-existing diagnostic test for an infection, but that defining test is only 85–90% accurate in diagnosing that infection, what does that say about the clinical utility of your mass spectrometry and machine learning system?

The COVID-19 pandemic has acutely highlighted the problems of validation of a test and the issues surrounding defining a clinical diagnosis on a test result alone with no other confirmation, including clinical symptoms, i.e., identifying who is or is not infected with the SARS-CoV-2 virus. In the case of rtPCR testing for SARS-CoV-2, this centres around the now widely used definition of “asymptomatic carriers” not as ever-developing symptoms but as just an rtPCR-positive qualitative result, and at whatever threshold of detection a test has specified in their emergency use authorization (EUA) submission. This has been taken to be the gold standard of defining a COVID-19 infection and against which all other tests are compared and validated.

However, comparative studies of rtPCR tests based on saliva versus nasopharyngeal swab sampling had to address this problem and, even with this self-defining of “asymptomatic COVID-19 infection”, they concluded that rtPCR had an overall clinical efficacy of 84.8% sensitivity and 98.9% specificity [38]. The realization is that, even amongst those patients who develop symptoms, the rtPCR test is not 100% sensitive. Indeed, one early study of the population prevalence of anti-SARS-CoV-2 IgG antibodies had found that in participants with a previous negative rtPCR test, 8% had developed IgG antibodies [39]. However, the issue of false positives and true asymptomatic individuals is a more significant one. Why all SARS-CoV-2 rtPCR test positives, who do not develop symptoms, are immediately classified as asymptomatic carrier, and at least some are not defined as clinical false positives, is a poorly debated point. The estimates of pre-symptomatic and asymptomatic SARS-CoV-2 infections from a wide variety of general population screening studies ranges from as low as 4% to more than 80%. However, studies that specifically differentiated pre-symptomatic from asymptomatic detection rarely estimated an asymptomatic fraction greater than 50% [40,41,42].

It is very unsafe to assume that there is not a false positive rate to rtPCR testing of clinical samples for SARS-CoV-2 infection, yet many studies do: Tran et al. describe the performance characteristics of an MLS, termed MILO, applied to MALDI–ToF mass spectra of 226 nasal swab extracts from individuals also tested for SARS-CoV-2 infection by rtPCR only. Of these, 107 were rtPCR-positive, and 159 were rtPCR-negative. The results show a 100% agreement with rt-PCR-positive samples and 96% agreement on rtPCR-negatives [20]. There are several fundamental false assumptions in this study that need to be addressed, as have already been highlighted earlier, but the machine learning issue is an apt example.

Assuming an 85% sensitivity and 5% false positive rate for the rtPCR tests on the clinical samples used, a reasonable estimate is that 10% of the samples in this study were misclassified by the rtPCR test. The biological markers measured in the MALDI-ToF mass spectra are proteins, whereas rtPCR is priming from viral RNA; these are different but correlating markers of the infection. Given that they measure different markers and that each method has inherent experimental errors, the probability of exact concordance in all samples is virtually zero. Thus, to report an accuracy as agreement with the rtPCR classification of greater than 80% indicates over fitting.

There is an often-repeated and very harsh saying in bioinformatics of “garbage in garbage out”. In clinical MALDI–ToF mass spectrometry, initially, a lot of this “garbage” contamination of data is due to inconsistent protocols, poor calibration, detector and sample-ionization drift. Addressing analytical problems methodically, many of these analytical issues can be solved by good laboratory science and practice. However, the final poorly recognized problem for machine learning and contamination of data is the correct classification of cases as categorically infected or not infected. Machine learning algorithms pivot on this one definition, and the system will be geared to correctly identifying all classified positives and negatives with 100% accuracy.

As clearly demonstrated during the COVID-19 pandemic, in reality, we do not have the ability to categorically say yes or no that a sample is from a person who is infected (or infective), only a probability that they are. Currently, the best probability tests are rtPCR systems, but that probability of being correctly classified varies and should NOT be absolutely defined by rtPCR “positivity” alone. Thus, any given machine learning system reporting near-perfect accuracy for identifying COVID-19, when the confidence that the sensitivity of the defining test to diagnosis a SARS-CoV-2 infection correctly (i.e., rtPCR) is at its very best 85%, is open to question. Indeed, if the MALDI-ToF mass spectrometry–machine learning system reported was to improve on rtPCR diagnosis of SARS-CoV-2 infection, you would not expect it to match the rtPCR result with such conformity. Therefore, greater confidence should be given to a trimmed ML algorithm matching at a level of 50–80% with an existing test, not one reporting near 100% concordance.

When attempting to achieve a machine learning algorithm with 98–100% accuracy, the confidence of YES/NO that the sample is from someone with and without the disease also has to be 98–100%. Few, if any, single diagnostic tests match this expectation. Thus, supervised machine learning needs to account for the level of uncertainty and probability of correct classification of real-world sample data sets [43,44].

## 3. Clinical Testing Laboratory, Validation and International Accreditation

Validation data are required for the authorization of a test to be used in clinical diagnostics [45]. This also has to consider the operational requirement of testing, i.e., within a category 3 laboratory, or category 2 laboratory facility after making the sample inert. Accreditation then has to consider the requirements of efficacy proof required for emergency use authorization (EUA), versus post-pandemic adoption into clinical practice via laboratory-developed test (LDT) legislation and then the much more rigorous requirements of FDA and IVDR legislation.

Although most tests for SARS-CoV-2 infection (COVID-19) have received EUA approval, this is not a good strategy for a viral screen test that has greater utility than just SARS-CoV-2 detection, and with a view to post-pandemic utility in seasonal viral infection screening and biosecurity, such as in rapid boarder/aviation testing of passengers.

Emergency use authorization (EUA) allows the U.S. Food and Drug Administration (FDA) and other nations’ regulatory bodies to help strengthen public health protections against CBRN threats. “CBRN” is the abbreviation commonly used to describe the malicious use of chemical, biological, radiological and nuclear materials or weapons with the intention to cause significant harm or disruption. By international agreement, EUAs, by facilitating the availability and use of medical counter measures, i.e., diagnosis and screening, may also be required and issued during public health emergencies arising from naturally occurring and emerging diseases, such as with COVID-19. However, such approval for the use of such tests could be withdrawn if such a state of emergency is suspended or terminated [46].

The U.S. Food and Drug Administration (FDA) recently clarified that, when it grants an emergency use authorization (EUA) for a point-of-care test, that test is deemed to be CLIA-waived. For the duration of the national emergency declaration for COVID-19, such tests can be performed in any patient care setting that operates under a CLIA Certificate of Waiver or Certificate of Compliance/Certificate of Accreditation. In addition, the FDA clarified that tests for SARS-CoV-2 that are offered prior to or without an EUA have not been reviewed by FDA, are not FDA authorized, and have not received a CLIA categorization external icon. Thus, those tests are considered of high complexity by default until they receive an EUA or other FDA approval that indicates they may be performed as moderate complexity or waived tests.

So, such allowance to use could be terminated, as they are not FDA-approved, and this is conditional on the level of the health threat to the public. The vast majority of COVID-19 tests are EUAs, which is not the same as being FDA-approved, and clinical use authorization will be removed at some point.

### Limitation and Advantages of CLIA-Laboratory Developed Tests (LDT)

The clinical laboratory improvement amendments (CLIA) regulate laboratory testing and require clinical laboratories to be certified by the Center for Medicare and Medicaid Services (CMS) before they can accept human samples for diagnostic testing [46]. Laboratories can obtain multiple types of CLIA certificates, based on the kinds of diagnostic tests they conduct. Three federal agencies are responsible for CLIA: The Food and Drug Administration (FDA), Centre for Medicaid Services (CMS), and the Centers for Disease Control and Prevention (CDC). Each agency has a unique role in assuring quality laboratory testing.

A laboratory-developed test (LDT) is a type of a diagnostic test that is designed, manufactured and used within a single CLIA-certified laboratory. LDTs can be used to measure or detect a wide variety of analytes (substances such as proteins) in a sample taken from a human body. Some LDTs are relatively simple tests that measure single analytes; others are complex and may measure or detect many analytes simultaneously such as by MALDI–ToF mass spectrometry [47]. The regulations and guidelines state that the use of LDTs is the same as using FDA-cleared or -approved in vitro diagnostic tests.

We validated our MALDI–ToF tests for virus infection by SARS-CoV-2 as an LDT high-complexity test governed by FDA CLIA guidelines in a CLIA Laboratory. It was validated and compared against an FDA EUA-approved SARS-CoV-2 rtPCR test. Such a validation will not have the same risk of possible suspension in the future as an EUA does but constrains use to CLIA laboratory use and internal re-validation at each CLIA laboratory adopting the MALDI–ToF mass spectrometry viral infection detection technique.

## 4. MALDI–ToF Mass Spectrometry Screening Test for SARS-CoV-2

The procedural protocol developed in the UK at the laboratories of Viral Zoonosis, University of Cambridge, and MAPSciences, The iLab, Bedford UK is fully described by Iles et al. [15]. This was reproduced and utilized at the University of Northern Illinois [48].

The basic protocol is:Prepare positive control: Take 300 μL of ultra-pure water and add to the freeze-dried positive control SARS-CoV-2 pseudo-virus tube Vortex until there are no visible clumps in the tube. Spike 150 μL into 4.85 mL of saline. Process as for samples.Prepare gargle-saliva samples: Irradiate collected gargle-saliva samples with UV light at 280 nm for 15 min Collect the liquid into a lure lock syringe attached with a 21 G, 1.5″ needle. Once the liquid has been collected into the syringe, create an air bubble to keep the liquid in place whilst removing the needle from the syringe. Add the plastic cover and dispose it into a sharps box.Apply the 0.45 μL filter onto syringe and push 5 mL of sample/spiked saline through a 0.45 μL filter into a clean 10 mL centrifuge tube.Viral particle enrichment: Add ice-cold acetone (4 °C) to match the amount of sample in the tube (1:1); it should be 5 mL, giving a total volume of 10 mL. Centrifuge and spin the samples at 16,000 RCF at 4 °C, for 30 min. Discard the supernatant completely into a suitable container. Note: No visible pellet can be observed.Dissolution of virions and solubilization of viral proteins: Add 50 μL of the buffer LBSD-X working solution made up with 20 mM TCEP. Manually, with the pipette tip, scrape it off the side of the container and pump mix with a pipette. Vortex the sample well, and incubate for 15 min at a room temperature.MALDI–ToF mass spectrometry: After 15 min of sample incubation with the buffer LBSD-X and TCEP, vortex the sample again to ensure homogeneity and that 1 μL of the sample is ready to plate following the sample plating procedures using 15 mg/mL Sinapinic Acid matrices.

Typical spectra generated will detect viral spike proteins and other viral envelope proteins (VEPs) along with salivary–mucosal IgA antibody light and heavy chains in a patient sample, as shown in Figure 8.

## 5. CLIA LDT Validation of a MALDI-ToF Mass Spectrometry Test for SARS-CoV-2

Gargle-saliva samples analysed on MALDI-ToF mass spectrometers in a CLIA laboratory, criteria for validation of laboratory developed test:Limit of detection (LoD);Interfering substances and specificity;Clinical performance evaluation;Reproducibility;Stability.

### 5.1. Limit of Detection (LoD)

Determined as a detection of S1, S2, S2′ peaks above a combined threshold, the lowest amount of analyte in a sample that can be detected needs to be established. The LoD is considered the lowest concentration that triggers a positive result 95% of the time. LoD is determined by diluting a sample with a known concentration of virus and testing until the assay no longer detects at a >95% positive rate (see Table 1).

In a biological assay of infectious SARS-CoV-2 (Wuhan isolate), the MALDI–ToF spectral signal was no longer detectable at 10 PFU/mL. A signal would be seen 95% of the time at about 700 PFU/mL.

In an rtPCR assay comparison, a MALDI-ToF spectral signal was no longer detectable for a sample corresponding to 77 copies/mL, as quantitated by rtPCR. A signal would be seen 95% of the time at about 400 copies/mL.

NOTE—Discrepancies exist between rtPCR tests in their estimation of copy numbers, and the serial dilution of culture media of the Wuhan isolate indicated that 1 PFU equated to 88 copies of SARS-CoV-2 genome in an in-house rtPCR. Non-functional fragment detection is often a cause of these discrepancies between functional and nucleic acid sequence detection assays.

### 5.2. Interfering Substances and Specificity

Interference of endogenous compounds that can be found in saliva were tested to evaluated interference with results of test. The following substances were tested for interference with the assay. Three pools of blank saliva samples and three pools of positive saliva samples were tested in replicates of six. No interference was defined as spiked blank samples testing negative and spiked positive samples testing positive. Interference was defined as blank samples testing positive and positive samples testing negative. In these experiments, a positive test was determined as the detection of Spike S1, S2 and S2′ mass spectral peaks characteristic of the SARS-CoV-2 virus (see Table 2).

### 5.3. Clinical Performance Evaluation

The intended purpose is mass population screening to identify pre-symptomatic and *asymptomatic carriers* of SARS-CoV-2.

Clinical evaluation was specific to this purpose.○No other test for SARS-CoV-2 testing with emergency use authorization has been evaluated for this purpose in the clinical performance evaluations.○All previous EUA IVD tests have been evaluated on hospitalized, clinically confirmed, patient samples only.

The asymptomatic are people who have caught the infectious agent, do not develop symptoms AND infect others who will develop the disease. Identifying these is a difficult process. Although the purpose of population screening is to identify pre-symptomatic and asymptomatic samples via a simple affordable test, the problem of false positives is often ignored by the need for near 100% sensitivity. Clinically, those that score positive on an rtPCR and do not develop symptoms are assumed to be asymptomatic with no other biomarker or clinical verification. Even by self-defining “asymptomatic infection”, a recent study concluded rtPCR had an overall clinical efficacy of 84.8% sensitivity [35]. However, the issue of clinical false positives was ignored. It is recognized that, on average, 30–50% of positive SARS-CoV-2 rtPCR test results identify individuals with no symptoms and who never developed COVID-19 symptoms, however mild. It is not demonstrated which of these pass on the infection. Estimates of asymptomatic SARS-CoV-2 infections from a wide variety of studies range from as low as 4% to more than 80%. However, studies that specifically differentiated pre-symptomatic from asymptomatic detection rarely estimated an asymptomatic fraction greater than 50% [40,41,42]. Nevertheless, despite evidence that lateral flow devices can detect viral antigens with an acceptable degree of reproducibility on mock samples, real-world testing indicates that the Innova lateral flow SARS-CoV-2 antigen device has only a 3–4% sensitivity for rtPCR-positive pre-symptomatic/asymptomatic samples when screening the general population [49]. Given that the rtPCR test is not a perfect gold standard upon which to base an evaluation, it is the best we have. Thus, although a strong correlation is going to be extremely unlikely for any alternate testing modality, the magnitude of discrepancy between the Innova lateral flow and nasopharyngeal swab rtPCR testing of a symptomless population for SARS-CoV-2 infection is alarming. Accepting that only broad comparisons can be made, in the case of the Innova test, the specificity is probably close to 99.9%, but sensitivity is very low for the purpose of identifying pre-symptomatic and asymptomatic carriers. Conversely COVID-19 rtPCR tests may have high sensitivity for asymptomatic, with possibly an 80–90% detection rate, but specificity at the minimum LoD thresholds (high Ct) is likely to be much lower but impossible to determine. The published data available indicate that the clinical specificity of high-Ct single-primer rtPCR SARS-CoV-2 could potentially be as low as 70% when applied to a symptomless population screening for pre-symptomatic/asymptomatic individuals spreading the virus.

In light of the above issues, the gargle-saliva MALDI–ToF test incorporated both signature mass spectral peaks detection characteristic of the SARS-CoV-2 virus and a marker of elevated immunological activity with the oral mucosa of the upper respiratory tract (i.e., IgA heavy-chain mass spectra peak measurement). This was then compared against general population screening in order to give an indication as to its ability to achieve non-invasive easy sampling mass population screening for pre-symptomatic and clinically significant asymptomatic individuals.

#### 5.3.1. Establishing Operational Cut-Off Values

A negative population was evaluated consisting of 150 samples 3 months after recovery from SARS-CoV-2 infection (123 medical staff mild COVID-19 symptoms) and from convalescent patients (27 ITU patients after discharge), all with confirmed serological-positive SARS-CoV-2 antibodies. None of the 150 exceeded the threshold of positive testing in the combined S1, S2, S2′ peak measure AND IgA heavy chain on the Shimadzu 8020 mass spectrometer.

In a second control population analysis of 300 members of the UK, the general population (volunteers from London and Bedford, UK) gave gargle-saliva samples. Eleven scored positive on the gargle-saliva MALDI–ToF test at a conservative threshold. When the positive thresholds were increased by a factor of two, 7 of the previous 11 scored positive. One individual (marked with f) reported having experienced COVID-19 symptoms requiring bed rest 2 weeks prior to giving a sample (see Figure 9).

The prevailing population estimates of infection amongst those NOT reporting symptoms for Bedfordshire and North London UK varied from 2.5% up to 3.5% (for London, UK) at the time of sample collection (November 2020 to January 2021—office of national statistic UK Gov). Thus, the expected rtPCR positive rate or pre-symptomatic and asymptomatic carriers was 8 to 11.

The extrapolated estimated detection of “pre-symptomatic and asymptomatic infection” therefore varied from 64–88% to 100–138% depending on the threshold applied of the expected rtPCR detection rate.

Note the UK government rtPCR SARS-CoV-2 testing conforms to a minimum threshold of two of three primer pair positives for a Ct of <35 to be recorded as rtPCR positive for COVID-19 infection.

#### 5.3.2. Direct Comparison with an rtPCR SARS-CoV-2 Test

As rtPCR is the mainstay of the global diagnosis and population screening of the SARS-CoV-2 pandemic; all other clinical testing technologies have to be compared against “PCR”-based assessments. The EUA-approved Abbott SARS2 rtPCR test was used as comparator in this evaluation.

The study group came from 550 student athlete samples at Northern Illinois University taken between November and December 2020. All were nasopharyngeal swab sample tested using the Abbott rtPCR SARS-2 test. Of these, 77 were rtPCR-positive, 10 of which developed symptoms. That is a positivity rate of 14%, of which 66 (85.7%) are defined as rtPCR “asymptomatic”. Thus, overall, amongst this population, 12% were rtPCR asymptomatic carriers [46].

One hundred and fifty-two gargle-saliva samples were collected simultaneously with naso-pharyngeal swabs. A comparative evaluation study was performed to evaluate the performance of the MALDI COVID-19/SARS-CoV-2 test detection of pre-symptomatic and asymptomatic using the Shimadzu Axima MALDI-ToF mass spectrometer. Of the 152 gargle-saliva samples analysed, three were pre-symptomatic rtPCR-positive and 57 rtPCR-positive, which determined asymptomatic; ninety-two were rtPCR-negative samples. In this evaluation, both virus S1, S2, S2′ spectral mass peaks combined threshold and that of the IgA heavy chains must be exceeded to score positive (see Table 3).

#### 5.3.3. Internal Sampling QC

In order to determine if a negative is a true negative and not due to poor sample collection, non-digestive (ND) mucus gel non-mucin structural proteins found in oral and respiratory mucus and saliva were monitored in all clinical samples. The spectral 10,900 to 11,900 m/z peaks correspond to ND non mucin proteins co-precipitating with MUC-glycoproteins during acetone extraction. Gastric and intestinal mucus contains only non-mucin mucus gel proteins smaller than 6KD [50]. They are only liberated from the viscous mucous gel complex by extensive disulphide reduction. The 10,900 to 11,900 m/z proteins are endogenous compounds of oral respiratory tract mucus and saliva. These peaks should always be present in the correctly collected gargle-saliva sample mass spectra. If a peak at 10,900 to 11,900 m/z peak is not present, then the sample should not be considered in a good condition, and the results are not valid.

#### 5.3.4. Daily System QCs

The positive control is SARS-CoV-2 pseudotype at 10^3^ FU/mL or heat-inactivated SARS-CoV-2, which is processed as if a saliva-gargle sample. Pooled human pre-2019 saliva is used as the negative control. A confirmed positive (positive control) and confirmed negative sample (negative control) must be run with each plate run on the MALDI-ToF test. The positive must be positive, and the negative sample must be negative for results of any given sample plate to be accepted. If either control is out, samples must be rerun.

### 5.4. Reproducibility

Reproducibility shows the ability of the assay to generate the same result from the same sample. The reproducibility of the assay is demonstrated by injecting a minimum of four positive samples and four negative samples for 4 days. Intraday reproducibility will be determined as result of a single day. Inter-day will be the total over the 4 days. Reproducibility is calculated as positive measured/positive expected × 100 and negative measured/negative expected × 100 (see Table 4). Over a four-day period, sample scores remained constant.

### 5.5. Stability

Storage sample stability determined by storing spiked samples (positive control CDC inactivated Virus and Pseudo-type virus) frozen at −18 to −20 °C, at same conditions patient samples will be stored and at time 0, 1 week, 2 weeks, 3 weeks, and 4 weeks.

Extract stability was determined by leaving samples on the MALDI–ToF sample plate after time 0 analysis and rerunning at 12 h, 24 h, and 48 h. Acceptance for stability of all stability samples is no change in result (see Table 5).

## 6. Conclusion—Validation Disposition

In this CLIA-LDT validation, we recommend that the test is used as a general screen of the population at large in order to identify pre-symptomatic and asymptomatic carriers of SARS-CoV-2 infection. Setting a low threshold confirmation by rt-PCR for those scoring positive is recommended for the increased probability of maximal asymptomatic carrier diagnosis.

Repeat regular testing of gargle-saliva samples is recommended for continued surveillance and follow-up.

The sample is a 30 s gargle/saliva. Mouthwash prior to sampling is an interfering substance to the process (as it disrupts the viral particle structure prior to size partition and concentration).

The cross-identification of coronavirus OC43 and 229E can occur. Confirmation of positives with rtPCR is recommended.

Rapid first screen by cheaper MALDI-ToF MS and home sampling will reduce costs and resources pressure on rtPCR services that are 10× higher.

### Conclusion—Global Biosecurity

The year 2020 marked the beginning of a new era in human viral infections, in which the coronavirus SARS-CoV-2 emerged as the defining contagious disease of the 21st century. It is going to be essential that we are able to detect viral-infected individuals and trace contacts if future outbreaks are to be successfully managed. As nations have now moved from lockdown to “back to work”, it is imperative that we are able to rapidly contain subsequent outbreaks of variants and new viral infection challenges.

MALDI–ToF mass spectrometry-based tests are going to play a significant role in future diagnostic surveillance and screening for viral outbreaks. The mass patterns resolved by this technique are specific to different viruses, virion particles, and proteomes. Bioinformatic analysis scores the pattern association with a known virus, but all are recorded. Thus, influenzas and other respiratory viruses are detected and characterized by this technique, but their pattern does not match, and it is therefore not recognized as SARS-CoV-2. More refined bioinformatics will not only identify that a virus infection is detected, but also what that virus is, within seconds [51]. In order to do so, multi-disciplinary research groups are required to achieve further advances in virion particle enrichment, mass spectral bioinformatics and MALDI-ToF instrument design to enable robust, high-throughput systems to be deployed [52,53].

## 7. Patents

RK Iles & JK Iles US 2021/0356475 A1: Virus and exosome sample preparation and analysis methods.

## Figures and Tables

**Figure 1 viruses-14-01958-f001:**
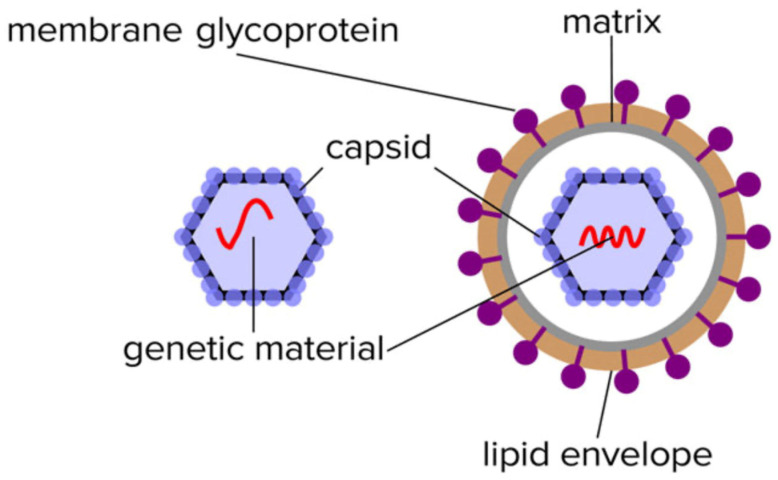
Illustration of the limited composition of unique proteins found in the isolated non-enveloped an enveloped virion particles.

**Figure 3 viruses-14-01958-f003:**
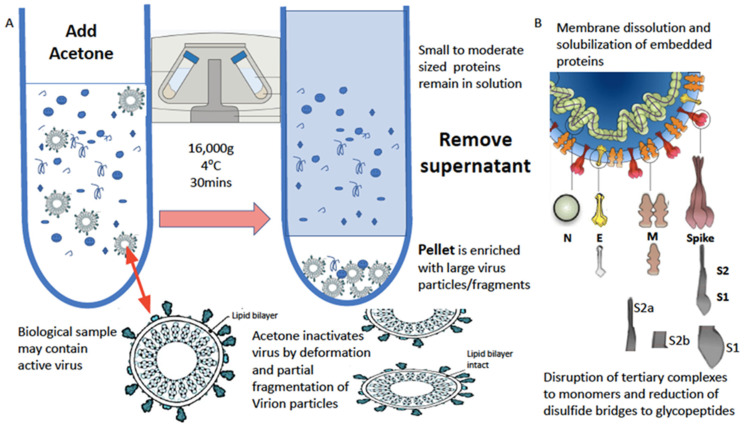
Process of virion enrichment within a biological sample and extraction/solubilization of Spike (S) and other viral envelope protein for MALDI ToF MS analysis. Panel (**A**): Biological samples containing virus are mixed 1 to 1 *v/v* with ice cold acetone and centrifuged at 16,000× *g* for 30 min at 4 °C. The supernatant containing smaller non-precipitated solutes is discarded. The pellet is enriched with viral particles and kept for analysis. Acetone treatment inactivates envelope virus by deformation and partial fragmentation of the viral envelope and embedded protein Structures A. Panel (**B**): The pellet is resuspended in 10 to 100 μL of MALDI-ToF mass spectrometry compatible dissolution and solubilization buffer. Termed LBSD-X this buffer does not suppress ionization and contains a detergent at a concentration optimized to release viral envelope embedded proteins together with non-embedded viral proteins. It also contains dithiothreitol (DTT) in order to further reduce di-sulphide bonds so that quaternary and tertial structures are fully disrupted and monomers, polypeptide chains and glyco-polypeptides are liberated for detailed mass analysis [15].

**Figure 4 viruses-14-01958-f004:**
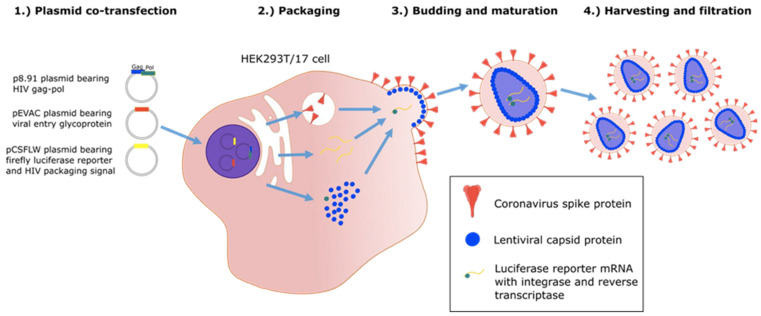
Graphic illustration of production of “pseudo-type virus” expressing envelope proteins such as coronavirus spike protein which are correctly post translational modified [24].

**Figure 5 viruses-14-01958-f005:**
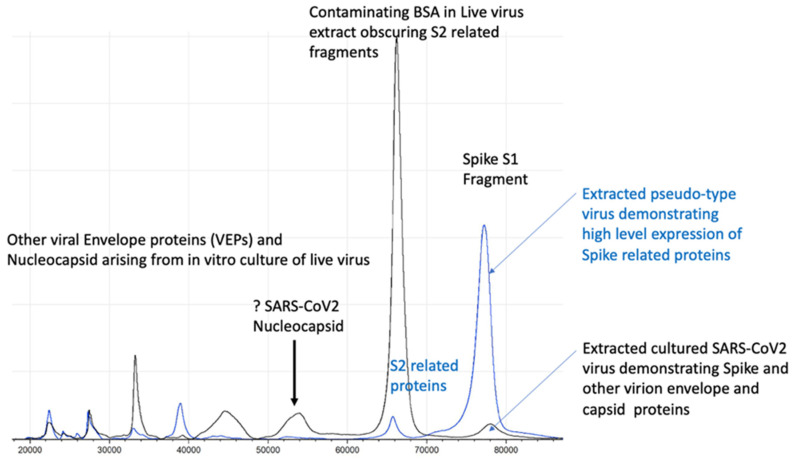
Illustration of full profile of virion proteins detected by MALDI–ToF not seen in pseudo-type virus.

**Figure 6 viruses-14-01958-f006:**
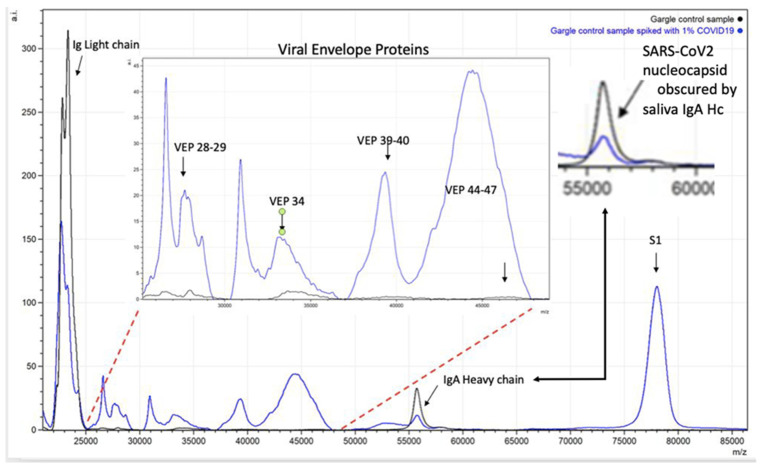
Spectra illustrating how endogenous sample proteins from gargle-saliva can hide key virion proteins (VEPs) such as nucleocapsid peaks by co-enrichment of endogenous IgA.

**Figure 7 viruses-14-01958-f007:**
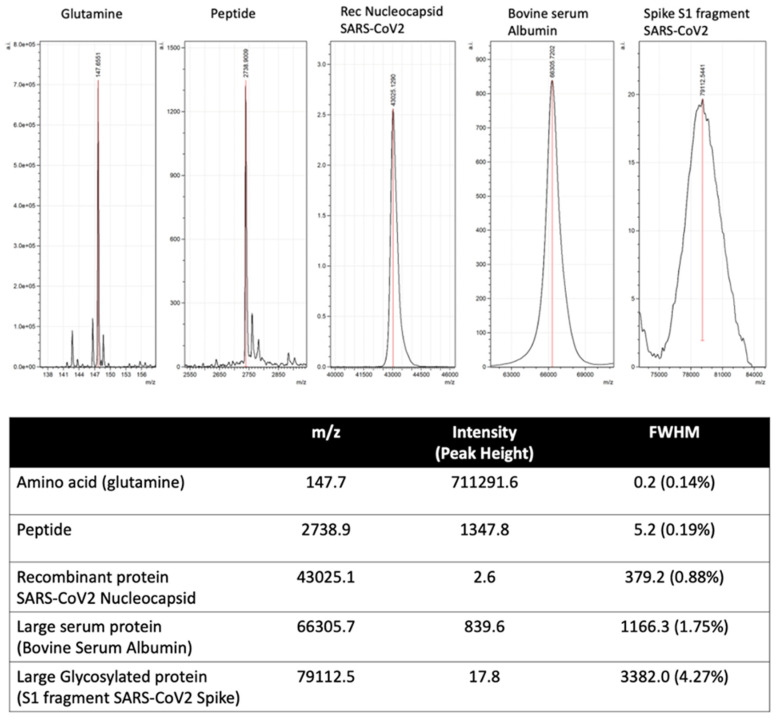
Illustration of mass spectral peak spreading due to increased complexity of combined isotopic composition and post-translational modification of large proteins.

**Figure 8 viruses-14-01958-f008:**
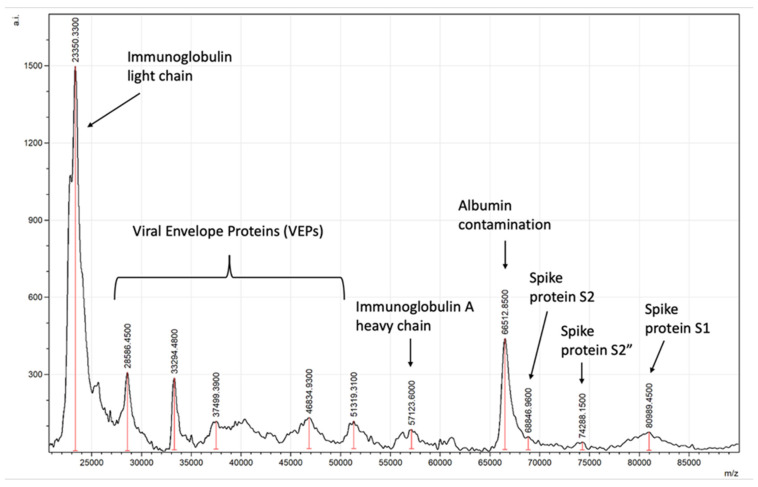
Example of asymptomatic SARS-CoV-2-positive saliva gargle sample mass spectra.

**Figure 9 viruses-14-01958-f009:**
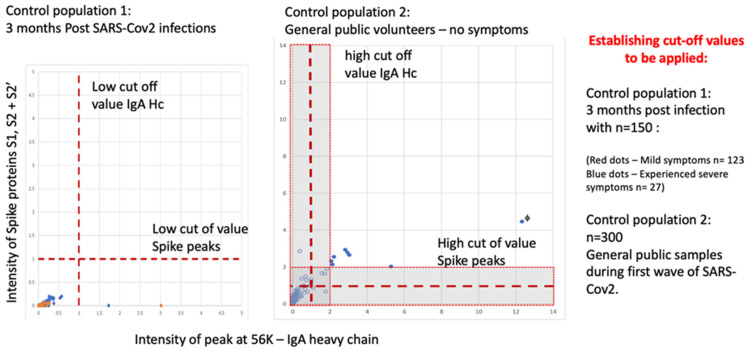
MALDI–ToF MS test incorporating IgA as a clinical feature in clinical patient samples (not seen in virus spiked samples) determining cut off values for virus spike protein and immune response IgA (heavy chain) combined.

**Table 1 viruses-14-01958-t001:** Tabulated dilutions of SAR-CoV-2 to determine the consistency of detection in the MALDI–ToF MS test and limit of detection (LoD), measured as: (**A**), biological plaque forming units (PFU) determined from serial dilution of in vitro culture media of SARS-CoV-2 (Wuhan) spiked into negative saliva-gargle; and (**B**), quantitative rtPCR determined copy number of SARS-CoV-2 from a positive sample dilutes into negative saliva-gargle.

**(A) PFU**		
**Conc.**	**POS/NEG**	**% Positive**
1000 PFU (200 per mL)	10/10	100%
750 PFU (150 per mL)	10/10	100%
500 PFU (100 per mL)	5/10	50%
10 PFU (2 per mL)	0/10	0%
**(B) Copy Number**		
**Conc.**	**POS/NEG**	**% Positive**
2304 copies per mL	10/10	100%
1152 copies per mL	10/10	100%
576 copies per mL	10/10	100%
288 copies per mL	9/10	90%
144 copies per mL	5/10	50%
77 copies per mL	0/10	0%

**Table 2 viruses-14-01958-t002:** Determination of interfering substance and cross reaction with other viruses in the MALDI–ToF MS test for SARS-CoV-2.

		Positive Samples Detected			Negative Samples Detected		
Potential Interfering Substance	Concentration of Substance	Pool 1	Pool 2	Pool 3	Pool 1	Pool 2	Pool 3
Toothpaste	NOT EVALUATED						
Mouthwash *	0.1% *v/v* Benzalkonium chloride And/or 0.1% Chlorohexidine	**neg**	**neg**	**neg**	**neg**	**neg**	**neg**
Nicotine	NOT EVALUATED						
Benzocaine/Menthol							
Caffeine							
No Interfering substance (controls)	Pseudotype SARCOV-2 10^3^ FU/μL	Pos	Pos	Pos	neg	neg	neg
SARS Virus	Pseudotypes at 10^3^ FU/μL	Pos	Pos	Pos	neg	neg	neg
Rhino Virus	Infected Individuals Gargle (conc^n^ unknown)	Pos	Pos	Pos	neg	neg	neg
Human Coronavirus MERS	Pseudotypes at 10^3^ FU/μL	Pos	Pos	Pos	neg	neg	neg
Human Coronavirus OC43 **	Pseudotypes at 10^3^ FU/μL	Pos	Pos	Pos	**Pos**	**Pos**	neg
Human Coronavirus 229E **	Pseudotypes at 10^3^ FU/μL	Pos	Pos	Pos	neg	neg	**Pos**
Human Coronavirus NL63	Pseudotypes at 10^3^ FU/μL	Pos	Pos	Pos	neg	neg	neg
Human Coronavirus HKU1	Pseudotypes at 10^3^ FU/μL	Pos	Pos	Pos	neg	neg	neg
Influenza A virus (H1N1 Swine Flu)	1000 PFU	Pos	Pos	Pos	neg	neg	neg
Influenza A virus (H1N21 Pueto Rico)	1000 PFU	Pos	Pos	Pos	neg	neg	neg

Functional infectivity assay determination of concentration: FU = Fluorescence Unit. This is a functional transfection measure of light detection from reporter luciferase gene being expressed as a result of pseudo type virus successfully infecting target HEK293 cells in vitro. FU is determined from serial dilution of the pseudo virus samples infection of HEK293 cells in vitro. PFU = Plaque-forming Units. A biological measure of functional pathogenic virus infected Vero cells from co-culture with live virus containing samples. PFU is determined from serial dilution of the virus samples plaque formation on a Vero cell in vitro culture monolayer. * Persons who are providing samples to be analysed by the test must abstain from mouthwash 1 h prior to sample collection. ** Phylogeny related beta-coronaviruses OC43 and alpha-coronavirus 229E have similar characteristic S1, S2 and S2′ mass peaks which can score positive on this evaluation.

**Table 3 viruses-14-01958-t003:** Tabulated results of a comparison of the MALDI–ToF MS test for SARS-CoV-2 and an EUA rtPCR test with a low threshold (**A**) and high threshold (**B**) of scoring positive the detection of elevated S1, S2 and S2′ viral proteins and IgA heavy chain in the saliva gargle sample.

**(A)** **High Cut-Off Threshold**	**Abbott SARS-CoV-2 rtPCR Test ****			
		**Pre**- **Symptomatic** **Positive (3)**	**Asymptomatic** **Positive (57)**	**Negative (92)**
**MALDI–ToF** **Gargle** **sample test**	Pre-Symptomatic Positive (3)	**3/3**	N/A	0
	Asymptomatic Positive (47)	N/A	**29/57**	18
	Negative (102)	0	28	**74/92**
**Positive** **Agreement**	Pre symptomatic cases 100% Asymptomatic cases 51%			
**Negative** **Agreement**	80% agreement on negatives			
**(B)** **Low Cut-Off Threshold**	**Abbott SARS-CoV-2 rtPCR Test ****			
		**Pre**- **Symptomatic** **Positive (3)**	**Asymptomatic** **Positive (57)**	**Negative (92)**
**MALDI–ToF** **Gargle** **sample test**	Pre-Symptomatic Positive (3)	**3/3**	N/A	0
	Asymptomatic Positive (47)	N/A	**45/57**	18
	Negative (102)	0	5	**102/92**
**Positive** **Agreement**	Pre symptomatic cases 100% Asymptomatic cases 75%			
**Negative** **Agreement**	65% agreement on negatives			

Overall agreement with the Abbott SARS2 rtPCR classification of positive or negative is 70%. Of the Abbots SARS2 rtPCR positives classification of samples this varied from 53% to 77% depending on thresholds applied. ** Note this EUA rtPCR test requires only one of two SARS-CoV-2 genome (RdRp and Nucleocapsid) sequence primers to be positive at <40 Ct to be recorded as qualitatively positive.

**Table 4 viruses-14-01958-t004:** Tabulated determination of repeat positive samples and repeat negative samples to be detected over 4 days.

	Week 1 POS/NEG	Week 2 POS/NEG	Week 3 POS/NEG	Week 4 POS/NEG	Total POS/NEG
Positive (PCR)	4/0	4/0	4/0	4/0	16/0
Negative (PCR)	0/4	0/4	0/4	0/4	0/16

**Table 5 viruses-14-01958-t005:** Composite tabulation of storage stability of saliva/gargle spiked samples containing inactivated SARS-CoV-2 virus (CDC inactivated virus (**A**,**C**)) and pseudotype expressing SARS-2 spike protein (**B**,**D**), as processed acetone precipitate “Extract” (**A**,**B**) and as complete frozen mock “Sample” (**C**,**D**).

**Extract Stability**					
**(A) CDC Inactivated** **Virus**	**Week 1** **POS/NEG**	**Week 2** **POS/NEG**	**Week 3** **POS/NEG**	**Week 4** **POS/NEG**	**Total** **POS/NEG**
Positive (PCR)	5/0	5/0	5/0	-	15/0
Negative (PCR)	0/5	0/5	0/5	-	0/15
**(B) Pseudo-Type Virus**	**Week 1** **POS/NEG**	**Week 2** **POS/NEG**	**Week 3** **POS/NEG**	**Week 4** **POS/NEG**	**Total** **POS/NEG**
Positive (PCR)	10/0	10/0	10/0	10/0	40/0
Negative (PCR)	0/10	0/10	0/10	0/10	0/40
**Sample Stability**					
**(C) CDC Inactivated** **Virus**	**Week 1** **POS/NEG**	**Week 2** **POS/NEG**	**Week 3** **POS/NEG**	**Week 4** **POS/NEG**	**Total** **POS/NEG**
Positive (PCR)	5/0	5/0	5/0	5/0	20/0
Negative (PCR)	0/5	0/5	0/5	0/5	0/20
**(D) Pseudo-Type Virus**	**Week 1** **POS/NEG**	**Week 2** **POS/NEG**	**Week 3** **POS/NEG**	**Week 4** **POS/NEG**	**Total** **POS/NEG**
Positive (PCR)	5/0	5/0	5/0	5/0	20/0
Negative (PCR)	0/5	0/5	0/5	0/5	0/20

## Data Availability

Not applicable.

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
