# Peer review of "A How to Guide: Clinical Population Test Development and Authorization of MALDI-ToF Mass Spectrometry-Based Screening Tests for Viral Infections"

_viruses, 2022, doi:10.3390/v14091958_

Round 1

Reviewer 1 Report

The authors, Iles et al, have presented a very comprehensive guide for application of MALD-ToF MS for clinical testing of viruses. The review is comprehensive with up-to-date work in the field. The review should be accepted for publication, after minor corrections to the text and figures.

1. There are a minor typos in the text and context of abbreviation for example under fig 1, chromatographic is abbreviated to LC, but elsewhere it is liquid chromatography. 

2. Figures 2 and 3 need citation of source from where this was adapted.

3. m/z ratio needs formatting throughout the text to italics as community standards.

4. It is not clear in Fig 7, what the question mark in label represents for SARS-CoV2 nucleocapsid. 

5. Inset in figure 8 is blurry and requires high resolution image. 

6. The tables inserted in section 5, have red lines under text. Instead of figures, the authors should consider inserting the original tables.

7. It is unclear from the conclusion how MALDI based tests are applicable beyond SARS-CoV2 for global biosecurity. Authors should consider elaborating the application cases if there are any on other virus particles. 

Author Response

We much appreciate the reviewer comments. Our responses are as follows

1. The paper has undergone significant revision and editing. 

2. Three new references have been added and refer to the original sources of the composite image: "Original composite figure adapted in parts from Depfenhart et al [16] Kupferschmidt and Cohen [17] and Mason [18]."

3. editing throughout has occurred

4. The question mark is indeed a suggestion that this peak represents nucleocapsid. 

5. The insert has been made sharp

6. All tables have been revised

7. The wider application to other viruses has been elaborated on and referenced: 

"The mass patterns resolved by this technique are specific to the different viruses, virion- particle, proteomes. Bioinformatic analysis scores the pattern association with a known virus but all are recorded. Thus, influenzas and other respiratory viruses are detected  and characterized by this technique but their pattern does not match and is therefore not recognized as , SARS-CoV2.  More refined bioinformatics will not only identify that a virus infection is detected, but also what that virus is, within seconds [51]."

Reviewer 2 Report

The authors describe in this review MOLDI- Toff method to detect SARS-CoV2 infection . They grouped the patients in asymtomatic and symthomatic. They underlined in detail the methods used to isolate the samples from infected patients.

In my opinion, the article is suitable to Viruses publication. However, major revisions are required.

Major Points.

Q1. The English grammar and structure must be improved. 

Q2. Described the material and methods in a specific section.

Q3. The table structure must be modified. The authors could use only white and grey colours and change them in size.

Author Response

We thank the referee for their constructive comments. Our responses are:

1. The paper has been substantially revised for English and clarity.

2. As this is a review paper we have kept the structure and not created a material and methods section as it spans so many studies. However, we equally considered this to be an "original article" but on discussion with the Editorial team settled on this being formatted as a review.

3. We have made substantial revision of figures and tables.